# Molecular benchmarks of a SARS-CoV-2 epidemic

Hakon Jonsson [1], Olafur T. Magnusson[1], Pall Melsted[1,2], Jonas Berglund[1], Arna B. Agustsdottir[1],
Berglind Eiríksdottir[1], Run Fridriksdottir[1], Elisabet Eir Garðarsdottir[1], Gudmundur Georgsson[1],
Olafia S. Gretarsdottir[3], Kjartan R. Guðmundsson[1], Thora Rosa Gunnarsdottir[3], Hannes Eggertsson [1],
Arnaldur Gylfason[1], Hilma Holm [1], Brynjar O. Jensson[1], Aslaug Jonasdottir[1], Frosti Jonsson[1],
Kamilla S. Josefsdottir[4], Marianna Thordardottir[4], Karl G. Kristinsson[3], Þórður Kristjánsson[1],
Droplaug N. Magnusdottir[1], Louise le Roux [1], Jona Saemundsdottir[1], Asgeir Sigurdsson[1],
Gudrun Sigmundsdottir[4], Gardar Sveinbjornsson[1], Solvi Rognvaldsson[1], Ogmundur Eiriksson [1],
Magnus Karl Magnusson [1], Kristin Eva Sveinsdottir[1], Maney Sveinsdottir[3], Emil Aron Thorarensen[1],
Bjarni Thorbjornsson[1], Arthur Löve[3], Gudmundur L. Norddahl[1], Ingileif Jonsdottir [1,3], Patrick Sulem [1],
Gisli Masson[1], Alma Moller[4], Thorolfur Gudnason[4], Mar Kristjansson[3], Agnar Helgason[1,5],
Daniel F. Gudbjartsson [1,2], Unnur Thorsteinsdottir[1,3] & Kari Stefansson [1,3✉]

A pressing concern in the SARS-CoV-2 epidemic and other viral outbreaks, is the extent to which the containment measures are halting the viral spread. A straightforward way to assess this is to tally the active cases and the recovered ones throughout the epidemic. Here, we show how epidemic control can be assessed with molecular information during a well characterized epidemic in Iceland. We demonstrate how the viral concentration decreased in those newly diagnosed as the epidemic transitioned from exponential growth phase to containment phase. The viral concentration in the cases identified in population screening decreased faster than in those symptomatic and considered at high risk and that were targeted by the healthcare system. The viral concentration persists in recovering individuals as we found that half of the cases are still positive after two weeks. We demonstrate that accumulation of mutations in SARS-CoV-2 genome can be exploited to track the rate of new viral generations throughout the different phases of the epidemic, where the accumulation of mutations decreases as the transmission rate decreases in the containment phase. Overall, the molecular signatures of SARS-CoV-2 infections contain valuable epidemiological information that can be used to assess the effectiveness of containment measures.

[1] deCODE genetics/Amgen, Inc., Reykjavik, Iceland. [2] School of Engineering and Natural Sciences, University of Iceland, Reykjavik, Iceland. [3] Landspitali University Hospital, Reykjavik, Iceland. [4] Directorate of health, Reykjavik, Iceland. [5] Department of Anthropology, University of Iceland, Reykjavik, Iceland. ✉email: kstefans@decode.is

The fact that 135 million individuals (as of 11 April 2021) have been infected by the SARS-CoV-2 virus worldwide[1] underscores the explosive spread of the COVID-19 pandemic in part due to the inadequacy of the global infrastructure to deal with highly contagious pathogens. The first case of COVID-19 was diagnosed in Iceland on February 28, the largest number of active infections was on April 4th and on May 5th the first phase of the epidemic was all but over (Fig. 1). The key containment measures used by the authorities were to screen widely for the virus, put those infected in isolation, and track their contacts who were subsequently put in quarantine. The screening for SARS-CoV-2 was done in three ways, (1) those targeted by the healthcare system with signs and symptoms of disease or coming back from high-risk areas such as the ski resorts of Italy and Austria, (2) those who accepted an invitation to free screening, and (3) a random sample from the population. The total number of cases identified in the screenings was 1813 as of June 24. Most of the cases could be traced to a preexisting case through contact tracing. In Iceland, patients with confirmed SARS-CoV-2 infection were put in isolation for at least 2 weeks, and to be declared recovered they had to be symptom-free for a week.

During a SARS-CoV-2 infection, the virions infiltrate the host's cells and start replication of their genomes and sub-genomic products[2], and the viral concentration increases rapidly after infection as assessed with qPCR[3–5]. Then after the onset of symptoms, the viral concentration starts to decrease[3–5]. The mutations carried by virions from the transmitting host are passed to the receiving host and can be used to track the spread of the virions[6–11]. This means that there is substantial epidemiological information in the viral concentration and sequence diversity of SARS-CoV-2 infection. Here, we screened 63,701 nasopharyngeal samples and sequenced the virus from nearly all 1813 confirmed cases in Iceland. The sequence data in conjunction with the viral concentration allowed us to determine whether this phase of the epidemic was on the rise or waning when samples were collected. Having means to determine where an epidemic such as COVID-19 is in its trajectory is useful for the management of containment efforts.

## Results

**RNA viral concentration as a yardstick in an epidemic.** The number of SARS-CoV-2 cases in Iceland grew rapidly until the end of March when the effect of measures to slow the spread of the virus had materialized (Fig. 1). A total of 1813 cases had tested positive as of June 24 with a qRT-PCR test from nasopharyngeal and oropharyngeal samples[12]. Of the 1813 cases, 1631 were diagnosed by the healthcare system while 182 were diagnosed through population screening (Fig. 2A).

We used two methods to assess viral concentration in tested individuals: the cycle threshold (CT) values to detect a PCR product from the qPCR test and the fraction of sequence reads that mapped to the SARS-CoV-2 reference from viral RNA sequencing. Both showed the concentration in newly diagnosed cases decreases as the effect of the containment measures set in.

The qRT-PCR test provides a measure of viral concentration; the higher the viral concentration the fewer cycles of PCR amplification are needed to detect the virus[13]. The number of PCR cycles required to detect the virus was available for 1667 of the individuals who tested positive in Iceland. The qRT-PCR experiments were performed at Landspitali University Hospital (LUH) (E-gene probe) and deCODE genetics (ORF1ab, S and N gene probes) with 238 samples being tested at both facilities. The measurements at the two sites were highly correlated (Pearson's correlation $\rho = 0.93$; 95% CI: 0.91–0.95; Supplementary Fig. 1A). For the subsequent analysis, we averaged the CT measurements per individual for the samples with measurements at both facilities. The number of PCR cycles needed to detect the virus increased by 0.44 per week (95% CI: 0.24–0.64), interestingly, the number of cycles needed to be increased faster in the population screened individuals than in the ones targeted by the healthcare system (Fig. 2B, 0.91 95% CI: 0.66–1.16 cycle per week and 0.30 95% CI:−0.10–0.51 cycle per week, respectively). The symptomatic individuals targeted by the healthcare system were probably closer to the peak of their viral concentration, whereas individuals identified in the population screening were likely recovering from an asymptomatic or mildly symptomatic infection. To assess whether we could replicate the temporal decay in the viral concentration during the recovery phase in samples from other populations, we reanalyzed the qRT-PCR data from a contained epidemic in Australia[11]. In agreement with the Icelandic data, we found that the viral concentration decreased (1.04 cycle per week 95% CI: 0.72–1.36) as the epidemic in Australia was contained (Supplementary Fig. 2).

To assess how long the viral concentration persists in individuals deemed recovered, we invited those who had tested positive and had finished mandatory isolation to be qRT-PCR tested again. A total of 1144 recovered individuals gave additional samples between 10 and 81 days after first testing positive (Fig. 3A). Of those who still tested positive on the second test, 263 had a third test at least 7 days after the second test. In total 1407 samples were tested of which 364 were positive (26%), 981 were negative (70%), and 62 undetermined (4%). Of the 1144 participants, 1136 had at least one determinable test result. The fraction of individuals who tested positive halved every 13.3 days (95% CI: 12.3–14.1 days, Fig. 3B). Including the undeterminable samples in the analysis had a minimal effect; the number of individuals not testing negative halves every 13.7 days (95% CI: 12.5–14.1 days, Supplementary Fig. 3).

We sequenced viral RNA from 1782 out of 1813 SARS-CoV-2 cases in Iceland and of those 1507 have at least 90% of the SARS-CoV-2 reference genome covered by five or more reads. It may be assumed that the fraction of reads mapping to the SARS-CoV-2 reference increases with the amount of viral RNA in the sample and therefore the viral concentration. The fraction of mapped reads decreased with time during the epidemic (4.6%, 95% CI:

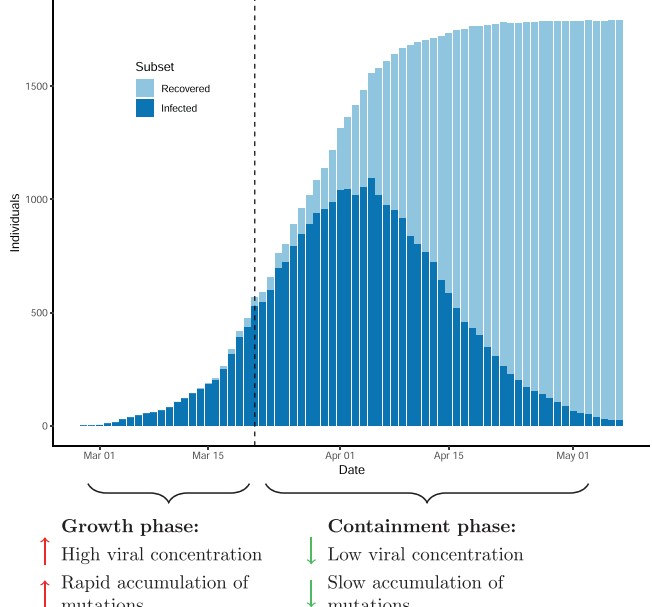

**Fig. 1 The molecular assessment of the Icelandic SARS-CoV-2 epidemic.** The number of infected and recovered individuals by date annotated with molecular markers of SARS-CoV-2 epidemic.

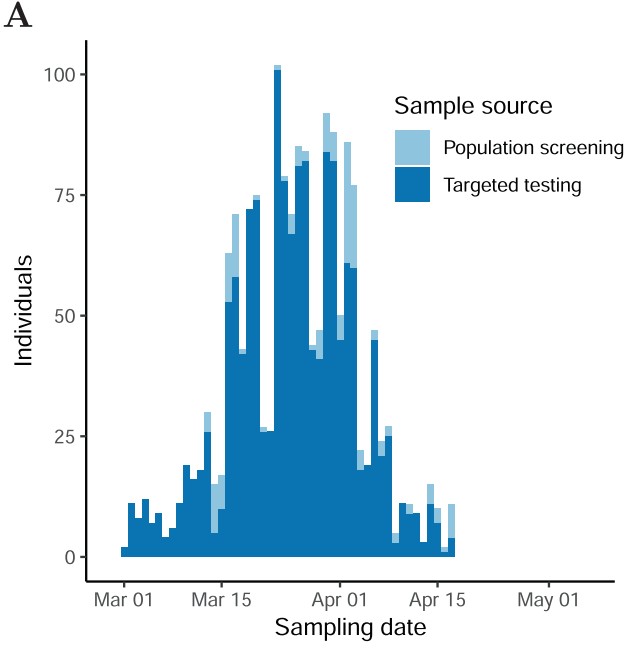

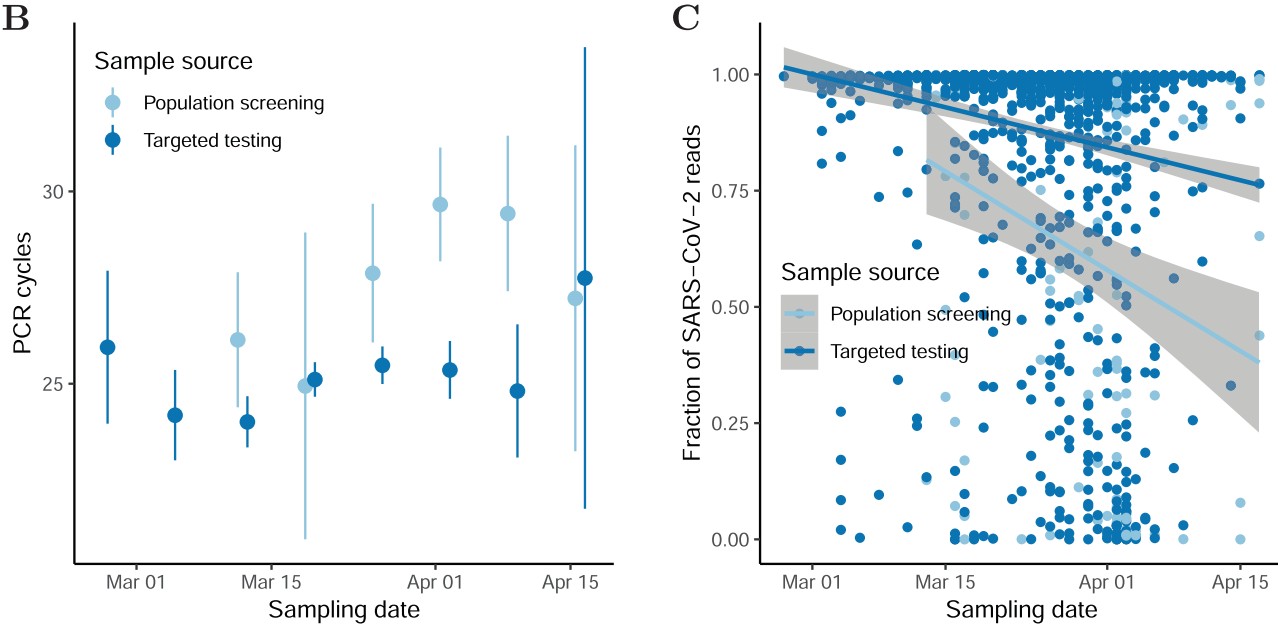

**Fig. 2 The Icelandic SARS-CoV-2 epidemic. A** Number of SARS-CoV-2-positive samples by time. **B** qPCR-CT values as a function of sampling date and sample source. The number of samples in the population screening bins is 45, 8, 32, 53, 11, and 8 ordered by date. The number of samples in the targeted screening bins is 34, 72, 202, 440, 454, 248, 53, and 5 ordered by date. **C** Fraction of SARS-CoV-2 reads mapping to reference as a function of sampling date and sample source. The center values in error bars and bands are means. The error bars and bands are 95% CIs.

5.7%–3.5% per week; Fig. 2C; Supplementary Fig. 1). This is in an agreement with the qRT-PCR results as the decrease indicates that individuals sampled later in the epidemic had lower viral concentration. The fraction of mapped reads decreased faster in the population screened individuals (9.0%, 95% CI: 10.3–7.6% per week) than the ones targeted by the healthcare system (3.5%, 95% CI: 4.6–2.4% per week).

**SARS-CoV-2 mutations provide a molecular signature of containment.** The rate of accumulation of mutations can be estimated by comparing the mutations that have accumulated in the sample from the root of the evolutionary tree of the virus to the date of the viral sample collection[14]. The accumulation rate of mutations is not only dependent on the mutation rate per viral replication but can also be affected by the number of virions, selection, and the length of the transmission interval.

The viral transmission consists of the initial population of virions from the transmitting host that proliferates in the receiving host. This transmission bottleneck is responsible for the fact that viruses primarily accumulate mutations when they are transmitted between people[15]. However, the possibility certainly exists that mutations that occur when the virus replicates within individuals may be detected if viral samples are taken from the same individual at multiple time points. Hence, we sequenced the virus from 203 retested individuals who

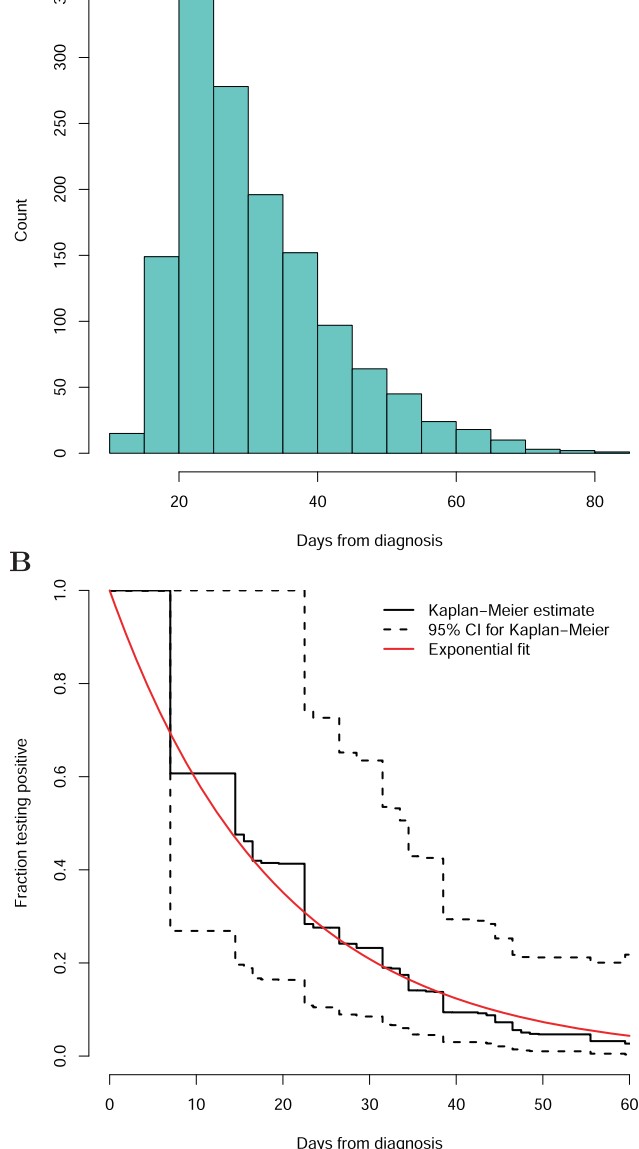

**Fig. 3 Molecular traces of SARS-CoV-2 recovery. A** The number of days separating the initial positive sample and the sample collected after recovery. **B** The fraction of individuals that remained positive as a function of days from the initial positive test.

had contracted COVID-19 and recovered from it. Only six of these samples (2.9%) yielded adequate sequence coverage of the SARS-CoV-2 genome. These six samples were taken >21 days after the initial sample and in none of them did we detect accumulation of mutations in the recovery phase, in agreement with results from Australia[11]. This suggests that the two main factors that could affect the rate of accumulation of mutations in our analysis are the length of time between viral transmissions and the time from infection to sample collection, both of which are likely to be longer when the epidemic is declining.

To estimate the accumulation rate of SARS-CoV-2 mutations per week, we regressed the number of mutations per viral genome on sampling date. The number of viral mutations that were found outside Iceland before 31st of March remained stable in the Icelandic population (0.05 mutation accumulated per week, 95% CI: 0–0.15), whereas the number of mutations that were rare in GISAID (fewer than 10 copies) accumulated at a rate of 0.18

mutations per week (95% CI: 0.12–0.23). This demonstrates that in Iceland, the accumulation of mutations was mostly due to local transmissions in Iceland. Interestingly, the accumulation rate in Iceland was lower than the accumulation rate derived from the public GISAID data (0.50 mutations per week; 95% CI: 0.48–0.52; Fig. 4A, B). Further, in Iceland, the rate of accumulation of mutations was substantially greater for the first 3 weeks (0.34 mutations per week, 95% CI: 0.18–0.47), than for the subsequent 4 weeks (0.10 mutations per week, 95% CI: 0.02–0.18; Fig. 4A). The slower rate of accumulation of mutations in the latter part of the epidemic in Iceland indicates that the generation time of infections was increasing or that SARS-CoV-2 cases had been infected for longer when first detected.

**Modeling the COVID-19 pandemic**. The higher CT values, the decreasing fraction of mapping sequence reads, and the reduction in the accumulation of mutations indicate that the individuals diagnosed late in the epidemic had been carrying the virus for longer than those diagnosed early, although we acknowledge that other scenarios are compatible with the data. As the viral generations of positive samples are unknown we modeled the containment in Iceland in an epidemiological model that tracks the viral generation of positive samples in Iceland and incorporates the observed molecular traces of SARS-CoV-2 infection. To model this prolonged detection of SARS-COV-2 viral RNA, we considered an extension of the susceptible, infected, and recovered (SIR) epidemiological model, which allows modeling of the spread of a disease in a closed population[16]. We extended this model to record the number of viral generations and included a state (SIPR model, Fig. 5A; Supplementary Fig. 4) with $P$ representing recovered individuals who still test positive for the virus but are not infectious, and thereby do not contribute to new viral transmissions. In this model, kappa controls the rate of transition from the positive ($P$) to recovered ($R$) states. We selected the following parameters for the SIPR model, initial susceptible population size ($S_0 = 350000$), initial infected population size ($I_0 = 100$), initial infection rate ($\alpha = 0.197/S_0$) and recovery rate ($\beta = \alpha/2$) that fit the Icelandic outbreak with an initial rapid growth that was then halted by containment measures ($\alpha_c = \alpha/10$). If the transition from a positive to a negative state is fast (kappa much >1) then the model is close to the SIR model. However, in the case of SARS-CoV-2, the rate of transition is slow (kappa much <1), the fraction of infectious individuals among positive individuals decreases rapidly after containment. Kappa mostly affects the observed accumulation of mutations after the containment measures, reflecting that samples from older viral generations still test positive (Fig. 5B). We next assessed the effect of the containment measures on the accumulation of mutations by varying the strength of the containment measure and fixing kappa at 0.5 (Fig. 5C). To reduce the accumulation of mutations by 30% within 28 days, 76% reduction in the number of viral generations is needed according to the SIPR model (black line in Fig. 5C). In other words, the observed decrease in the mutation accumulation rate as the epidemic progressed suggests the Icelandic measures decreased the rate of new viral generations by an order of magnitude.

## Discussion

One of the difficult challenges in managing containment efforts in an epidemic such as COVID-19 is to figure out at specific points in time where the epidemic is in its trajectory, is gaining or waning? In the work presented here, we show how changes in the viral concentration and in the accumulation of viral mutations can be used to determine the phase of the epidemic instead of simply counting new cases.

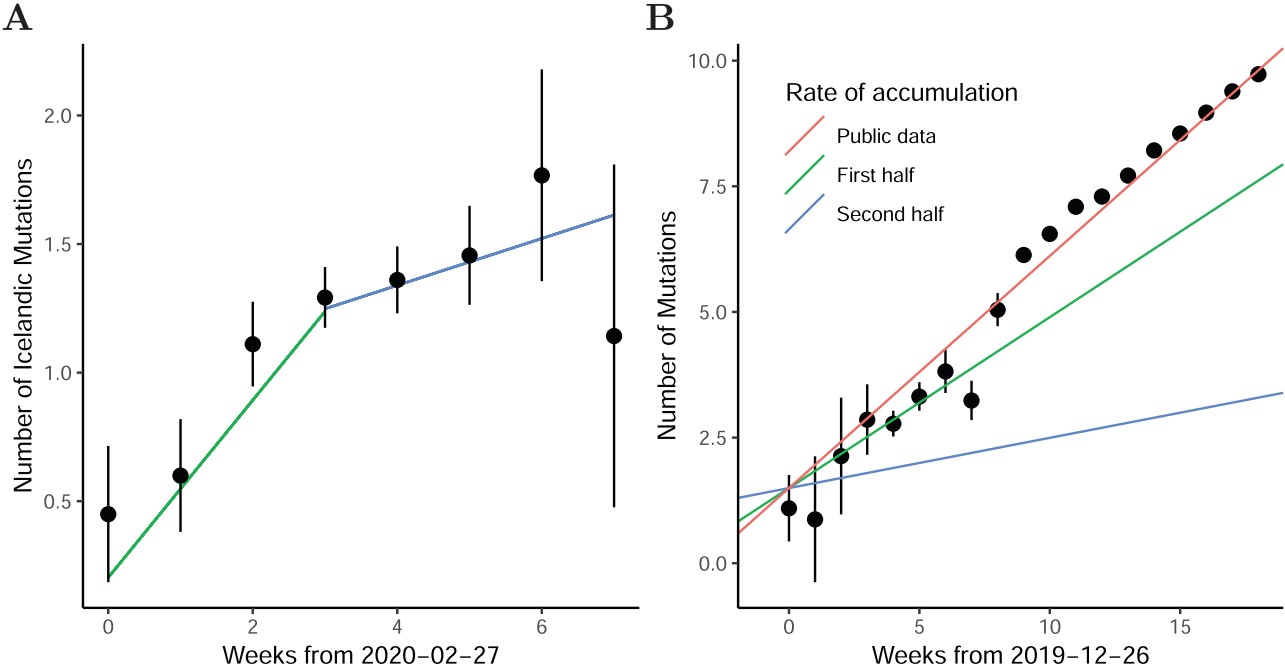

**Fig. 4 Accumulation of SARS-CoV-2 mutations. A** Number of Icelandic private mutations as a function of the sampling date. The number of samples in the bins is 20, 55, 225, 400, 388, 171, 43, and 7 ordered by date. **B** Number of mutations per sample that differ from the SARS-CoV-2 reference genome in the GISAID data as a function of the sampling date. The number of samples in the bins are 21, 8, 15, 85, 262, 226, 149, 171, 239, 1112, 2420, 4699, 6394, 8168, 6611, 4917, 4332, 3462, and 2676 ordered by date. The error bars are 95% CIs and the center values are means.

We estimated the viral concentration in two ways, with both methods we show how the viral concentration decreases as the epidemic wanes. However, whether the molecular traces of SARS-CoV-2 infections detected in samples with high CT values represent active virions or molecular remnants, is an open question. In Iceland, the individuals classified as recovered who still tested positive and had high CT values can shed some light on this. After 364 of these recovered individuals left isolation, we did not see a resurgence of COVID-19 cases indicating that these individuals did not have sufficient viral concentration to infect others and highlight the importance of prioritizing containment measures of COVID-19 cases based on the viral concentration. Viruses mainly accumulate mutations when they are transmitted between people. Hence, the accumulation of mutations can be used to assess the rate of transmissions. Furthermore, by estimating the rate of accumulation of mutations during the SARS-CoV-2 epidemic in Iceland, we demonstrate that in a rapidly spreading viral epidemic, the time from infection to sample collection is shorter than in an epidemic that has been slowed down and the time between transmissions is longer. This demonstrates that the mutation accumulation rate per week gives information about how fast the epidemic is spreading and can provide a quantitative assessment of the effectiveness of the measures taken to slow down the spread of the virus. In this interpretation, we assume that the time within an individual from initial infection to being infectious stays the same throughout the epidemic.

Our results suggest that the accumulation of mutations per week can be modeled as the product of the number of transmissions per week and the mutation accumulation rate per transmission. In our analysis, we have assumed the accumulation of mutations per transmission is constant throughout the epidemic. Stricter procedures in the containment phase could decrease the number of virions in the initial population of infection, which in turn could increase the probability of mutated virions reaching fixation in the next host, resulting in a higher mutation accumulation per transmission in the containment

phase. Therefore, future research is warranted in estimating the effect of containment on the accumulation of mutation per transmission.

We quantified the change in the viral concentration and accumulation of mutations as the epidemic progressed. We believe that standardization of molecular markers across laboratories would enable inter-cohort comparison to better estimate the current status of an outbreak rather than waiting for changes within the cohort to occur. The correlation between the reduced accumulation of mutations and the reduced viral concentration could be used to standardize viral concentration values across cohorts, by comparing the viral concentration of samples between cohorts with similar numbers of mutations.

## Methods

**Ethics statement**. The study was approved by the National Bioethics Committee of Iceland (approval no. VSN-20-070). Samples were taken from individuals after receiving informed consent from them or their guardians.

**Population screening and targeted testing**. We applied three previously described approaches[7] to detect SARS-CoV-2 infected individuals, briefly by targeted testing of individuals traveling from high-risk areas or been exposed to an infected individual; population screening of individuals, by offering people SARS-CoV-2 test free of charge and through sending an invitation by text message to a random subset.

**Sample collection**. For SARS-CoV-2 testing, nasopharyngeal and oropharyngeal samples were taken using collection swabs, either Sigma VCM$^{TM}$ Duo ENT or Sigma Transwab® ENT. RNA from all samples was isolated within 24 h.

**RNA extraction**. Viral RNA samples were extracted either at the Department of Clinical Microbiology laboratory at Landspitali, the National University Hospital of Iceland (LUH), or at deCODE. Both extraction methods are based on an automated magnetic bead-purification procedure, which includes cell lysis and Proteinase K treatment. RNA from samples at LUH were extracted (32 samples per 60 min run) using the MagNA Pure LC 2.0 or MagNA Pure Compact instruments from Roche LifeScience, with 200/100 μL input/output volume(s), respectively. Samples at deCODE were extracted from swabs (96 samples per 70 min run) using the Chemagic Viral RNA kit on the Chemagic360 instrument from Perkin Elmer,

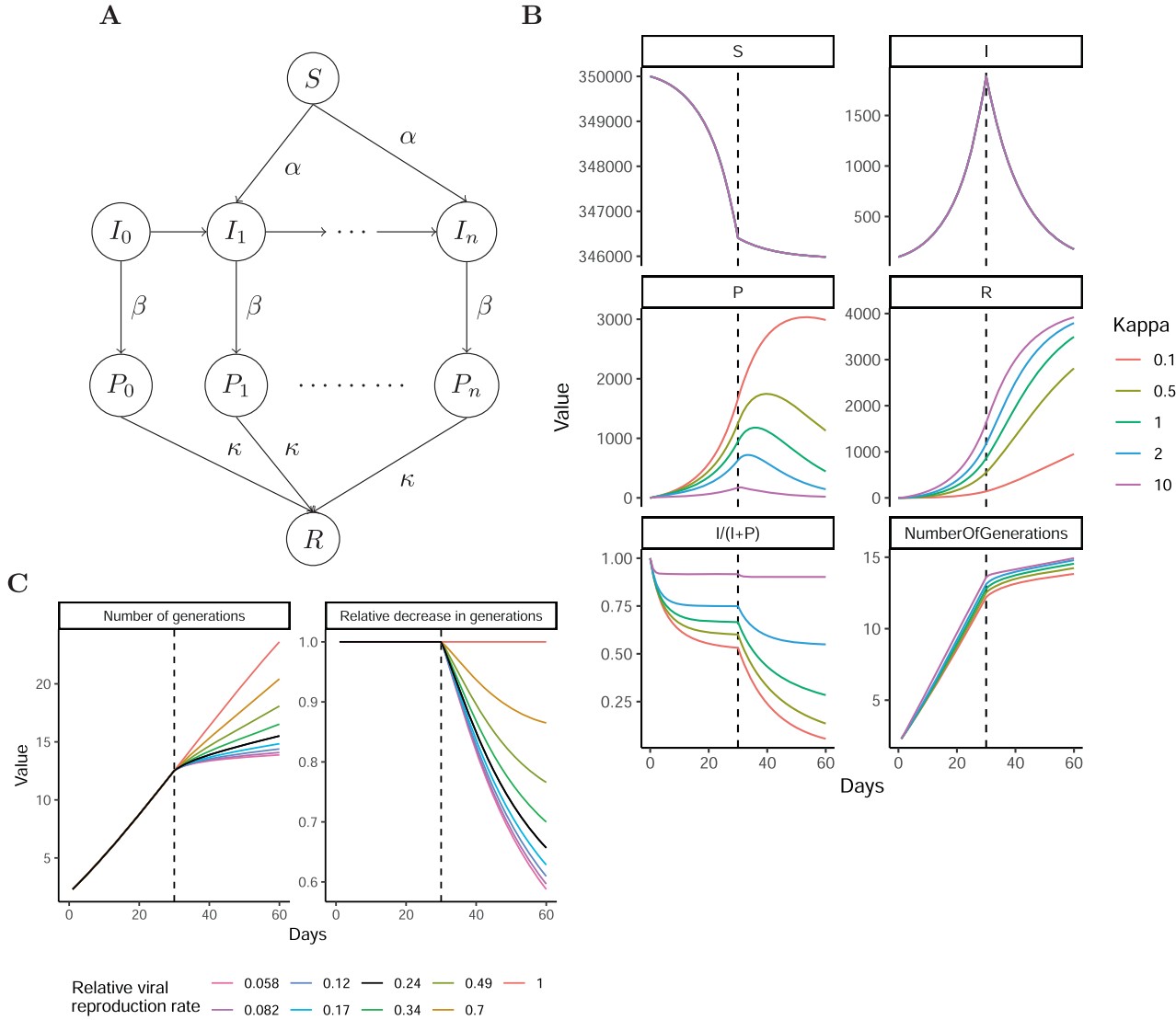

**Fig. 5 Susceptible, infected, positive, and recovered epidemiological model. A** schematic view of the flow of individuals in the epidemiological model. The susceptible individuals are infected by individuals in different infection generations ($I_0 \dots I_n$). These individuals then recover from the symptoms but test positive ($P_0 \dots P_n$). In the last stage, the positive individual's transition to the recovered and negative state ($R$). See Supplementary Fig. 4. **B** Number of individuals in each state $S$, $I$, $P$, and $R$ by time. $I$ and $P$ represent the aggregation of the generation states. The bottom panel portrays the number of viral generations by time. **C** The effect of containment measures on the number of viral generations.

with 300/100 µL input/output volume(s), respectively. Each step in the workflow was monitored using an in-house LIMS (VirLab) with 2D barcoding (Greiner, 300 µL tubes) of all extracted samples. MS2 bacteriophage RNA was added (2.5% vol) for all samples subsequently assayed with Method 2 (see below). All extracted RNA samples were stored at −80°C until further use.

**Testing of samples for SARS-COV-2 using qRT-PCR.** Testing for SARS-CoV-2 was performed either at LUH or deCODE using similar quantitative real-time reverse transcription PCR (qRT-PCR) methods. The assay at LUH is based on the WHO-recommended screening method (https://www.who.int/docs/default-source/coronaviruse/wuhan-virus-assay-v1991527e5122341d99287a1b17c111902.pdf), which involves a single probe pan-screening assay for betacoronaviruses, followed by confirmatory measurements for all positive samples using a nCoV-2019-specific assay. The broad betacoronavirus assay is based on probes for a conserved region of the E-gene, whereas confirmatory testing assays were done using either nCoV-2019 specific probes for the RdRp gene or the TaqMan™ Fast Virus 1-step Master Mix, 2019-nCoV Assay kits v1 from Thermo Fisher (see Method 1 below for details). All labeled probes and primers for the E-and -RdRP genes were from TAG (Copenhagen, Denmark). Superscript™ III One-Step RT-PCR assay mix with Platinum™ *Taq* DNA polymerase was from Thermo Fisher. 2019 E-gene control and SARS-CoV Frankfurt 1 positive controls were obtained from EVAg (https://www.european-virus-archive.com/bundle/diagnostics-controls-wuhan-

coronavirus-2019-2019-ncov). Each assay was done in a 25 µL total sample volume with FAM™ dye-labeled probes in addition to VIC™ dye-labeled probes for human RNase P as an internal control. Plates (96 well) were scanned in an AB-7500 Fast real-time PCR thermocycler for 40 cycles of amplification following the manufacturer's instructions (Thermo Fisher). Samples in the E-gene screening assay with $C_t < 35$ were considered positive and went for confirmatory testing using RdRp, whereas samples with $C_t$ values between 35 and 37 were considered weak positive and were confirmed using the TaqMan™ Fast Virus method. Samples with $C_t$ values from 37 to 40 were classified as inconclusive and were tested again to confirm their status.

SARS-CoV-2 screening at deCODE was performed using qRT-PCR assays in either a singleplex (Method 1) or a multiplex (Method 2) format, respectively. Method 1 uses the three probe TaqMan™ Fast Virus 1-step Master Mix, 2019-nCoV Assay kits v1, and 2019-nCov control kit from Thermo Fisher. Assay mix A, B, and C were prepared to contain FAM™ dye-labeled probes for the SARS-CoV-2-specific genes ORF1ab, S-gene, and N-gene, respectively. In addition, each assay mix contained VIC™ dye-labeled probes for human RNase P as internal control. Samples from 96-well RNA sample plate(s) were dispensed into three wells each in a 384 plate layout, in addition to three negative (no template) and three positive controls. Assay mix was added in a total reaction volume of 12.5 µL per sample. All sample aliquoting and mixing were performed with an automated Hamilton STARlet 8-channel liquid handler and the assay plates were scanned in an ABI 7900 HT RT-PCR system following the manufacturer's instructions with a total of

40 cycles of amplification. Samples with FAM^TM dye $C_t$ values <37 in at least two of three assays were classified as positive. Samples with FAM^TM dye $C_t$ values between 37 and 40 were classified as inconclusive and their testing repeated. If repeated testing gave the same result with at least two probes the sample was classified as positive. If repeated testing gave positive results for only one probe the test was considered inconclusive and a new sample from the subject was requested. The frequency of inconclusive results was 0.04%. Samples with undetected FAM^TM dye $C_t$ values or values equal to 40 in all three assays were classified as negative if the human RNase P assay was positive (VIC^TM dye $C_t < 40$). The sensitivity of the assay was evaluated by serial dilution of the positive control and was estimated at 6 copies per reaction (Supplementary Table 1).

Method 2 uses the TaqPath^TM COVID-19 CE-IVD RT-PCR kit from Thermo Fisher (Catalog# A48067). TaqPath^TM COVID-19 is a multiplexed assay that contains three primers/probe sets specific to different SARS-CoV-2 genomic regions and primers/probes for bacteriophage MS2 as a control, all measured in the same reaction well. The SARS-CoV-2 targets are the same as in Method 1, however, each target and the MS2 control contain probes with different reporter dyes (ORF1ab, FAM^TM; N-gene, VIC^TM; S-gene, ABY^TM; MS2, JUN^TM). The MS2 was used as an internal control and was added to each viral swab sample to assess the efficacy of the sample preparation. All reactions were performed in a 384-well plate layout in a total reaction volume of 6.25 μL per sample using an eight-channel Hamilton STARlet liquid handler for dispensing and mixing. A negative control (no template) and positive control (COVID-19 Control, 25 copies/reaction) were run on each plate. Assay plates were scanned in an Applied Biosystems^TM 7500 Real-Time PCR instrument following the manufacturer's instruction to a total of 40 cycles of amplification. Result criteria were the same as described for Method 1 above.

**Sample preparation for sequencing**. We used reverse transcription (RT) and multiplex PCR to generate cDNA from the RNA samples, we based our setup on information provided by the Artic Network initiative (https://artic.network/). Briefly, we pre-incubated extracted RNA at 65°C for 5 min in the presence of random hexamers (2.5 μM) and dNTP's (500 μM), we then cooled the sample. After cooling we performed RT using SuperScript IV (Thermo Fisher) in the presence of DTT (5 mM) and RNaseOUT inhibitor (Thermo Fisher) for 10 min at 42°C, followed by 10 min at 70°C. We next amplified the resulting cDNA using a multiplex PCR with a tiling scheme of primers. This scheme was designed to generate overlapping amplicons of approximately 800 bp (Supplementary Data 1). We split the primer pairs into two primer pools A and B, we then performed PCR reaction per sample and primer pool (Supplementary Data 1). We performed PCR amplification using the Q5® Hot Start High-Fidelity polymerase (New England Biolabs) with primers at 1 μM concentration. We performed the thermal cycles in an MJR machine with a heated lid at 105°C, using 35 cycles of denaturation (15 s at 98°C) and annealing/extension (5 min at 65°C). We then purified the resulting PCR amplicons using Ampure XP magnetic beads (Beckman Coulter). Further, we quantified the purified PCR amplicons using the Quant-iT^TM PicoGreen dsDNA assay kit (Thermo Fisher). We next randomly sheared the amplified samples (20–500 ng) to construct sequence libraries. The shearing was performed by focused acoustics in 96-well AFA-TUBE-TPX plates (Covaris Inc.) on the Covaris LE220plus machine with the parameters: sample volume, 50 μL; temperature, 10°C; peak incident power, 200 W; duty factor, 25%; cycles per burst, 50; time, 350 s. We used the fragmented cDNA to construct sequencing libraries in 96-well Covaris plates, using the NEBNext® Ultra II kit (New England Biolabs) following the manufacturer's instructions. Briefly, we performed end repair and A-tailing in a combined reaction per sample (plate) for 30 min at 20°C, followed by thermal enzyme inactivation at 65°C for 30 min. We next ligated adaptors to the cDNA templates using the NEBNext® ligation master mix plus enhancer and the TruSeq unique dual indexed IDT adaptors (Illumina, Supplementary Data 2). We incubated the ligation reactions for 15 min at 20°C. We purified the ligated sequencing libraries using a Hamilton STAR NGS liquid handler, with two rounds of magnetic SPRI bead purification (0.7× volume).

**ILLUMINA sequencing**. We pooled sequencing libraries (24–48 samples/pool) and quantified the pools using the Qubit dsDNA assay (Thermo Fisher). Depending on DNA concentration samples were diluted and denatured to a final loading concentration of 10 pM. We sequenced the pools using Illumina MiSeq sequencers with 300-cycle MiSeq v2 reagent kits (Illumina). We used dual indexed paired-end sequencing, this resulted in 150×8×8×150 bp cycles of data acquisition and imaging. We performed basecalling in real-time using MCS v3.1 and generated FASTQ files using MiSeq Reporter. For each run, we collected at least 15 M PE reads (>4.5 Gb) with base qualities of >Q30 for at least 90% of bases.

**Viral concentration regression**. We regressed the $C_t$ values on sampling date using the lm function in R. For replication, we fetched the Australian data set at https://github.com/MDU-PHL/COVID19-paper/raw/master/VIC.csv, and regressed the $C_t$ values on the sampling date as with the Icelandic data. To be comparable to the Icelandic data, we considered a subset where we excluded cases with $C_t$ values >37 and were detected prior to 2020/02/24. This had minimal impact on the slope estimate, as the slope estimate was 1.04 cycle per week (95% CI: 0.72–1.36) for the subset and 1.04 cycle per week (95% CI: 0.72–1.36) for the full set.

**Alignment and variant calling**. We aligned the sequences against the SARS-CoV-2 reference genome (NC_045512.2)[17] using bwa mem[18]. We marked possible PCR duplicate with mark duplicates from Picard tools (version 1.117) and we excluded reads with less than 50 bases aligned to reference from alignment. We called sequence variants and genotyped them jointly across all the samples with a modified version of Graphtyper[19] and the following settings:

  -no_filter_on_read_bias,-no_filter_on_strand_bias -no_filter_on_coverage -impurity_threshold=1.0 -primer_bedpe = {primer_file} -is_only_cigar_discovery -genotype_aln_min_support_ratio=0.30 -genotype_aln_min_support=5 -is_discovery_only_for_paired_reads -no_filter_on_begin_pos.

The modified version excluded the termini of reads overlapping a primer region to avoid primer and template chimeras. To call a mutation per sample, we required that Graphtyper called the alternative allele of the sequence variant in the sample. Further, we required that 80% of the reads supported the alternative allele and at least five reads supported the alternative or reference allele. If the sample did not meet these criteria, we flagged the mutation per sample as failed. If the failed fraction of samples per sequence variant exceeded 30% we did not consider the sequence variant for further analysis.

**Accumulation rate per time**. We dichotomized the mutations into two categories based on whether they were observed in the filtered GISAID data in 10 or more copies and the sample carrying the mutation in GISAID was sampled before the 31st of March. The accession numbers of the filtered GISAID set are in Supplementary Data 3. Per mutation subset, we regressed the number of mutations per sample as a function of the sampling date. We modeled this with a generalized linear model using Poisson and identity link. The modeling was implemented with the glm function in R[20]. We restricted to the samples with coverage across most of the genome, i.e., we excluded samples if the depth of coverage was strictly <5 reads for 500 genomic sites after restricting to bases of quality 20 or higher.

**Analysis of public data sets**. A total of 11,133 of 17,039 SARS-CoV-2 sequences available at the GISAID website on May 7th were used to estimate the accumulation of mutations per unit of time. Only almost complete and high-quality sequences were used in these analyses, where the first position was ≤200 and the last position was ≥29,750, where the no >2% of nucleotides were reported as totally ambiguous (N), no more than two positions were reported as partially ambiguous (Y, R, K, M, S, W, B, D, H, V) and no more than five deletions were reported. The sequences from Icelandic hosts, previously deposited by deCODE Genetics, were also excluded to ensure independence between the results based on deCODE sequences and those submitted to GISAID.

**SIPR model**. We constructed an epidemiological model using the ordinary differential equations depicted in Supplementary Fig. 4. For the numerical solution of the ordinary differential equations, we used the R-package deSolve. We solved the differential equation using two epochs, one with an exponential growth phase and then a containment phase. The S, I, P, and R values at the end of the exponential growth phase were used as initial values in the containment phase. In the exponential growth phase, the following parameters were used in the SIPR model: initial susceptible population size ($S_0 = 350000$), initial infected population size ($I_0 = 100$), infection rate ($\alpha = 0.197/ S_0$), and recovery rate ($\beta = \alpha/2$). Then in the containment phase, the alpha was decreased to model containment measures.

**Reporting summary**. Further information on research design is available in the Nature Research Reporting Summary linked to this article.

# Data availability

The SARS-CoV-2 sequences used in this manuscript are available at GISAID and ENA (PRJEB44803), the GISAID accession numbers are the following (with EPI_ISL prefix): 417481, 417535–417876, 424367–424624, 1585943–1585977, 1585979–1586097, 1586099–1586110, 1586112–1586121, 1586123, 1586125–1586178, 1586180–1586225, 1586227–1586267, 1586269–1586389, 1586391–1586416, 1586418–1586433, 1586435–1586438, 1586440–1586460, 1586463–1586490, 1586493–1586494, 1586496–1586571, 1586573–1586574, 1586576–1586589, 1586591–1586626, 1586628–1586636, 1586638–1586646, 1586648–1586662, 1586664–1586669, 1586671–1586777, 1586780–1586791, 1586793–1586809, 1586811–1586827, 1586829–1586846, 1586848–1586862, 1586864–1586883, 1586885–1586893. The consensus sequences used for the comparative analysis are available at GISAID, the accession numbers are supplied in Supplementary Data 3. Source data are provided with this paper.

# Code availability

Bwa mem was used to align the short-read sequences against the SARS-CoV-2 reference. Graphtyper was used to call sequence variants and for genotyping (version 2.3.0). Both are publicly available. The code to fit the ordinary differential equations from the epidemiological model is at https://github.com/hakon-jon/SARS-CoV-2-epi-model along with regression examples.

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

## Acknowledgements
We thank all the participants that donated samples.

## Author contributions
H.J., D.F.G., U.T., K.S. designed the study. H.J., O.T.M., P.M., B.E., R.F., E.E.G., O.S.G., T.R.G., H.H., B.O.E., A.J., F.J., K.S.J., M.T., K.G.K., D.N.M., L.R., J.S., A.S., G.S., G.S.v., S.R., K.E.S., M.S., A.L., G.L.N., I.J., P.S., G.M., A.M., T.G., M.K., A.H., D.F.G., and U.T. gathered samples/information. H.J., P.M., G.G., K.R.G., H.E., A.G., Þ.K., E.A.T., B.T., and G.M. wrote software for this study. H.J., O.T.M., P.M., J.B., H.E., A.G., G.S.v., O.E., P.S., A.H., D.F.G., and U.T. analyzed data. H.J., O.T.M., A.B.A., A.J., D.N.M., L.R., J.S., A.S., G.S., S.R., K.E.S., M.S., G.L.N., A.H., and D.F.G. did experiments. H.J., D.F.G., U.T., and K.S. wrote the manuscript with feedback from all the co-authors.

## Competing interests
H.J., O.T.M., P.M., J.B., A.B.A., B.E., R.F., E.E.G., G.G., K.R.G., H.E., A.G., H.H., B.O.E., A.J., F.J., Þ.K., D.N.M., L.R., J.S., A.S., G.Sv., S.R., O.E., M.K.M., K.E.S., E.A.T., B.T., G.L.N.,I.J., P.S., G.M., A.H., D.F.G., U.T., and K.S. are employed by deCODE genetics/Amgen, Inc., Reykjavik, Iceland. All other authors declare no competing interests.
