## [Peer Review File · Nature Communications]

REVIEWER COMMENTS

Reviewer #1 (Remarks to the Author):

In this study, 1813 cases were sequenced. The major findings of this study are:

- They have found that during the containment phase, the viral load is lower and there is a slower accumulation of mutations than those during the growth phase.
- The duration of viral shedding can be prolonged
- They have demonstrated that mutation rate is higher among unique mutations in Iceland. So, these mutations are likely to have arisen in Iceland or just before arriving at Iceland, and are related to local transmission.

The strengths of this study are:

- Original findings regarding the relationship between viral load/mutation rate and the period of the epidemic. These findings are especially of interest to evolution biologists.
- Large sample size
- The methodology is appropriate and the analyses are rigorous.

The weaknesses of this study are:

- The duration of viral shedding has been reported by many studies already.
- The author suggested that changes in viral load and accumulation of viral mutations can be used to determine the phase of the epidemic. However the phase of the epidemic can be simply determined by the number of cases. Therefore, the practical utility of these "markers" is limited.

Other comments:

Line 87-89: The number of PCR cycles cannot indicate the time from infection to viral sampling.

Line 100: How often did the patients have samples collected?

Reviewer #2 (Remarks to the Author):

In the present manuscript, the authors describe data and conclusions that they obtained from molecular investigations during the SARS-CoV-2 outbreak in Iceland. Differences in the detected virus concentration in symptomatic patients compared to asymptomatic individuals are presented. From the frequency of mutations in the virus genome, the authors conclude on the speed of transmission in the population and the effectiveness of containment measures.

The manuscript is written well and coherently, the presentation of the data is straightforward. Results and conclusions of the authors are definitely of interest and provide important information for molecular epidemiological investigations in connection with the COVID-19 pandemic. However, the following comments, suggestions and remarks should be considered before publication:

The authors use the term viral load throughout the manuscript. This term, however, originally comes from HIV therapy, where for the first time the determination of the concentration was decisive for the further course of therapy and initially just the "load" of the cellular immunity with infectious viruses was meant. Over time, the term was also partly applied to the hepatitis B virus and hepatitis C virus. In the meantime, the term viral load is wrongly used for any blood concentration of viral pathogens, the term virus concentration is more correct.

Essential statements of the authors are based, among other things, on differences in the number of PCR cycles required for virus detection. However, the CT values evaluated for this purpose are subject to significant method-specific variations and depend, among other things, on the respective genomic target, the PCR reagents or the PCR equipment used. A more detailed description of the laboratory methods used for molecular detection (if necessary as an appendix) would therefore be desirable for the critical reader.

If absolute genome quantification data (copies per milliliter) are available, they should be supplemented in the data sets.

In addition, possible pre-analytical factors, e.g. due to different sample collection techniques or sample transport times, should be discussed. These may have had an influence on the shifts in CT

values during the study period described by the authors.

When interpreting the significance of virus concentration and sequence differences in the virus genome, it should also be taken into account that the transmission bottleneck is not constant, but rather a complex function of both viral and host factors in the spread of SARS-CoV-2 in a population. Other influencing factors not covered by the authors' study should therefore also be taken into account.

Reviewer #3 (Remarks to the Author):

This paper presents a very interesting analysis of the molecular evolution of the COVID-19 epidemic in Iceland, both with respect to viral load of collected samples and molecular evolution inferred from the viral sequences from those samples. It is presented in a clear and concise format, and it is rich in stimulating data and analyses.

I found it a stimulating reading and I do not think I will be the only reader to find it so. Hence I recommend it for publication.

The technical details look fine to me.

However, I also found a few unmotivated statements and interpretations that should be properly rephrased or corrected before publication.

In line 87, "The number of PCR cycles needed to detect the virus increased by 0.44 per week ... indicating that the time from infection to viral sampling increased during the study period" is not justified by data.

It could be explained by a lower initial viral load, or a slower growth rate within host, or milder cases, or other effects.

Line 144: "The slower rate of accumulation of mutations in the latter part of the epidemic in Iceland indicates that the generation time of infections was increasing or that SARS-CoV-2 cases had been infected for longer when first detected." How would you exclude instead that this is due to a difference e.g. in the median viral load and therefore the intra-host population size of the virus, with consequent changes in e.g. selection pressure or fixation rates?

Or a different size of the bottleneck?

Line 150: "The higher CT values, the decreasing fraction of mapping sequence reads, and the reduction in the accumulation of mutations, all indicate that the individuals diagnosed late in the epidemic had been carrying the virus for longer than those diagnosed early" is one of the possible interpretations, but by far not the only one.

Do you have direct epidemiological support for that? Otherwise, please make it clear that it is a conjecture.

I find the epidemiological model and the discussion about viral generations quite confusing. From what I understand from Supp Figure 4, the viral generations refer to the number of transmissions of a viral lineage since introduction in Iceland. Is it right? Then please clarify.

But then, the "76% reduction in the number of viral reproductions" (line 169) refers a slow down of a factor of 4 in the times between viral generations, I guess? I.e. an impressively longer generation time. Could you comment on this? Is it consistent with epidemiological data?

Also, "viral reproductions" looks like it refers to within host replication, so please reword and clarify the whole discussion lines 166-172.

Please also adjust the claims in the Discussion according to the comments above.

Minor comments:

line 10: "that" were targeted by the healthcare system

Reviewer #1 (Remarks to the Author):

In this study, 1813 cases were sequenced. The major findings of this study are:

- They have found that during the containment phase, the viral load is lower and there is a slower accumulation of mutations than those during the growth phase.
- The duration of viral shedding can be prolonged
- They have demonstrated that mutation rate is higher among unique mutations in Iceland. So, these mutations are likely to have arisen in Iceland or just before arriving at Iceland, and are related to local transmission.

The strengths of this study are:

- Original findings regarding the relationship between viral load/mutation rate and the period of the epidemic. These findings are especially of interest to evolution biologists.
- Large sample size
- The methodology is appropriate and the analyses are rigorous.

The weaknesses of this study are:

- The duration of viral shedding has been reported by many studies already.
- The author suggested that changes in viral load and accumulation of viral mutations can be used to determine the phase of the epidemic. However the phase of the epidemic can be simply determined by the number of cases. Therefore, the practical utility of these "markers" is limited.

The prolonged viral concentration of SARS-CoV-2 has been reported by others and we describe that in the introduction. We are not claiming that as a novelty, what is novel in our study is bridging a gap between the usage of the molecular markers and epidemiological surveillance. We disagree with the reviewer that the utility of these markers is limited, the best demonstration of the practical value of these usage of the viral concentration by Icelandic government to assess the ongoing second wave in Iceland and to prioritize contact tracing of SARS-CoV-2 cases. For example, a low number of confirmed cases could be the remnants of an infection wave or recently imported active cases. The latter requires a nimble response by contact tracing teams or other measures to halt the spread. Therefore, if you have a mixture of cases with high and low viral concentration cases with limited contact tracing resources, you want to prioritize the contact tracing to the cases with high viral concentration.

Other comments:

Line 87-89: The number of PCR cycles cannot indicate the time from infection to viral

sampling.

We disagree, it has been described on an individual basis the viral concentration increases initially and slowly decays with time as the reviewer pointed out in the general comments about the manuscript. We recruited individuals leaving quarantine and tested them again. We find that the half time of testing positive is two weeks, or in another words for half of the positive individuals their viral concentration had decreased below the detection limit. This directly demonstrates that the time from the infection to sampling affects the viral concentration in the sample.

We are not stating we can predict for a particular individual when the individual was infected with any certainty, our main message is that we can discern between a recent active infections and a recovered individuals on the population level, which is of practical value for the containment effort.

We realize that there are possible alternative explanations to the observed reduction in viral concentration with time, however, in conjunction with the reduction in mutation accumulation we believe our interpretation is correct but perhaps premature at this place in the manuscript. We removed this interpretation from the sentence.

Line 100: How often did the patients have samples collected?

Individuals that tested positive for SARS-CoV-2 (1,813) were tested at least once. We invited individuals to be tested when they left isolation. 1144 individuals participated in this post-isolation testing and the number of post-isolation tests are in the following table.

Number of post-isolation tests	1	2	3	4	5
Number of individuals	930	175	30	8	1

Reviewer #2 (Remarks to the Author):

In the present manuscript, the authors describe data and conclusions that they obtained from molecular investigations during the SARS-CoV-2 outbreak in Iceland. Differences in the detected virus concentration in symptomatic patients compared to asymptomatic individuals are presented. From the frequency of mutations in the virus genome, the authors conclude on the speed of transmission in the population and the effectiveness of

containment measures.

The manuscript is written well and coherently, the presentation of the data is straightforward. Results and conclusions of the authors are definitely of interest and provide important information for molecular epidemiological investigations in connection with the COVID-19 pandemic.

However, the following comments, suggestions and remarks should be considered before publication:

The authors use the term viral load throughout the manuscript. This term, however, originally comes from HIV therapy, where for the first time the determination of the concentration was decisive for the further course of therapy and initially just the "load " of the cellular immunity with infectious viruses was meant. Over time, the term was also partly applied to the hepatitis B virus and hepatitis C virus. In the meantime, the term viral load is wrongly used for any blood concentration of viral pathogens, the term virus concentration is more correct.

We replaced the term viral load with viral concentration throughout the manuscript, we thank the reviewer for pointing this out and for the description of the historical context of the viral load term.

Essential statements of the authors are based, among other things, on differences in the number of PCR cycles required for virus detection. However, the CT values evaluated for this purpose are subject to significant method-specific variations and depend, among other things, on the respective genomic target, the PCR reagents or the PCR equipment used. A more detailed description of the laboratory methods used for molecular detection (if necessary as an appendix) would therefore be desirable for the critical reader.

We described in detail the qPCR protocols to the best of our ability in the initial submission in the methods. More specifically, we described the recruitment of individuals (Methods section: Population screening and targeted testing) and the swabs used for the sample collection (Methods section: Sample collection). Further we describe the two laboratories that the samples were processed at (Methods section: RNA extraction). The RNA extraction methods used at both locations (Methods section: RNA extraction) and the subsequent qRT-PCR assays and the definition of a positive samples (Methods section: Testing of samples for SARS-COV-2 using qRT-PCR).

We agree methodological differences could potentially hinder inter-cohorts comparisons on the absolute scale, however, we demonstrate the same relative decrease within our cohort with time with both the sequencing and qPCR approach. Further, we demonstrate similar relative decrease with time using the data from a contained wave in Australia. Despite the relative nature of our work we recognize the difficulty to harmonize the viral quantification across different laboratories as pointed out by the reviewer. To highlight this we added a discussion paragraph describing that dissemination of viral

concentration of positive cases along with technical description of the qPCR/sequencing could aid international effort to quantify the spread of SARS-CoV-2 and ultimately contain the COVID-19 disease.

“We quantified the change in the viral concentration and accumulation of mutations as the epidemic progressed. We believe that standardization of molecular markers across laboratories would enable inter-cohort comparison to better estimate the current status of an outbreak rather than waiting for changes within the cohort to occur.”

If absolute genome quantification data (copies per milliliter) are available, they should be supplemented in the data sets.

We do not have absolute quantification data for the samples tested. However, we performed dilution experiments to evaluate our experimental setup by serial dilution of control sample provided by ThermoFisher in the TaqPath kit. We added supplementary Table 4 describing the dilution experiments.

In addition, possible pre-analytical factors, e.g. due to different sample collection techniques or sample transport times, should be discussed. These may have had an influence on the shifts in CT values during the study period described by the authors. When interpreting the significance of virus concentration and sequence differences in the virus genome, it should also be taken into account that the transmission bottleneck is not constant, but rather a complex function of both viral and host factors in the spread of SARS-CoV-2 in a population. Other influencing factors not covered by the authors' study should therefore also be taken into account.

We added caveats to the interpretation of the CT values and describe the importance of calibration across laboratories in the discussion and how sequencing can aid in the standardization of CT values. We also point out in the same paragraph that in absence of calibration the relative CT within a laboratory are of value as relative increase or decrease would point toward containment of the virus.

“We quantified the change in the viral concentration and accumulation of mutations as the epidemic progressed. We believe that standardization of molecular markers across laboratories would enable inter-cohort comparison to better estimate the current status of an outbreak rather than waiting for changes within the cohort to occur. The correlation between the reduced accumulation of mutations and the reduced viral concentration could be used to standardize viral concentration values across cohorts, by comparing the viral concentration of samples between cohorts with similar number of mutations.”

We added to the discussion a paragraph describing that the transmission bottleneck is a function of the epidemic dynamics and we emphasize that further research is warranted in the characterizing this bottleneck as it could affect the proliferation of the transmitted viral population in the receiving host.

"Our results suggests that the accumulation of mutations per week can be modeled as the product of the number of transmissions per week and the mutation accumulation rate per transmission. In our analysis we have assumed the accumulation of mutations per transmission is constant throughout the epidemic. Stricter procedures in the containment phase could decrease the number of virions in the initial population of infection which in turn could increase the probability of mutated virions reaching fixation in the next host, resulting in a higher mutation accumulation per transmission in the containment phase. Therefore we acknowledge future research is warranted in the estimating the effect of containment on the accumulation of mutation per transmission."

Reviewer #3 (Remarks to the Author):

This paper presents a very interesting analysis of the molecular evolution of the COVID-19 epidemic in Iceland, both with respect to viral load of collected samples and molecular evolution inferred from the viral sequences from those samples. It is presented in a clear and concise format, and it is rich in stimulating data and analyses.

I found it a stimulating reading and I do not think I will be the only reader to find it so. Hence I recommend it for publication.

The technical details look fine to me.

However, I also found a few unmotivated statements and interpretations that should be properly rephrased or corrected before publication.

In line 87, "The number of PCR cycles needed to detect the virus increased by 0.44 per week ... indicating that the time from infection to viral sampling increased during the study period" is not justified by data.

It could be explained by a lower initial viral load, or a slower growth rate within host, or milder cases, or other effects.

Reviewer 1 raised similar concerns regarding the interpretation of the decrease in viral concentration with time. This interpretation is before the presentation of the reduction in accumulation of mutations with time as the epidemic was contained. The joint observation of decrease in viral concentration and accumulation of mutation in the containment, supports our interpretation that in the containment phase we are sampling

people later after infection compared to the growth phase. We agree that at this point in the manuscript the interpretation is perhaps premature.

We removed the interpretation from this sentence:

"The number of PCR cycles needed to detect the virus increased by 0.44 per week (95% CI: 0.24-0.64)"

Line 144: "The slower rate of accumulation of mutations in the latter part of the epidemic in Iceland indicates that the generation time of infections was increasing or that SARS-CoV-2 cases had been infected for longer when first detected." How would you exclude instead that this is due to a difference e.g. in the median viral load and therefore the intra-host population size of the virus, with consequent changes in e.g. selection pressure or fixation rates?
Or a different size of the bottleneck?

To clarify the answer, we will refer to the census and effective population size of the virus within a host, i.e. the number of virions and the number of unique viral haplotypes in these virions. Note that the number of viral haplotypes is less than the number of virions. We see decrease in the viral concentration with time, indicating that on average the census population size is decreasing with the time as the epidemic progresses. However, how this exactly relates to effective population size of the virions is unclear. If the containment of the epidemic would drastically decrease the effective population size of the virus, then accumulation of mutations would be expected to increase at a faster pace per transmission, as there would be post infection bottleneck which would increase the probability of mutations reaching fixation during infection. It is possible that the accumulation rate of mutations per transmission would increase as the epidemic is contained as the reviewer points out perhaps due to changes in hygiene protocols. However what we are measuring is the accumulation of mutations per time unit rather per transmission. We see slower rate in the accumulation of the mutations per time unit in the latter half of the first wave, which indicates that fewer number of generations per time in containment is stronger than its effect of on accumulation of mutation per transmission.

Line 150: 'The higher CT values, the decreasing fraction of mapping sequence reads, and the reduction in the accumulation of mutations, all indicate that the individuals diagnosed late in the epidemic had been carrying the virus for longer than those diagnosed early' is one of the possible interpretations, but by far not the only one. Do you have direct epidemiological support for that? Otherwise, please make it clear that it is a conjecture.

This is a conjecture and in response to that we have added a qualifier to the sentence to make it clear that this is one of many possible interpretations of the results.

"The higher CT values, the decreasing fraction of mapping sequence reads, and the reduction in the accumulation of mutations, indicate that the individuals diagnosed late in the epidemic had been carrying the virus for longer than those diagnosed early, although we acknowledge that other scenarios are compatible with the data."

I find the epidemiological model and the discussion about viral generations quite confusing. From what I understand from Supp Figure 4, the viral generations refer to the number of transmissions of a viral lineage since introduction in Iceland. Is it right? Then please clarify.

But then, the "76% reduction in the number of viral reproductions" (line 169) refers a slow down of a factor of 4 in the times between viral generations, I guess? I.e. an impressively longer generation time. Could you comment on this? Is it consistent with epidemiological data?

Yes, we were referring to the number of viral generations within Iceland or in other words the length of the transmission chain from the onset of the epidemic in Iceland to the collection of the positive sample.

We added sentences to the paragraph describing the motivation behind our modeling, which is to assess the number of generations from SARS-CoV-2 entering Iceland to sampling of the positive sample.

"As the viral generations of positive samples are unknown we modeled the containment in Iceland in an epidemiological model which tracks the viral generation of positive samples in Iceland and incorporates the observed molecular traces of SARS-CoV-2 infection."

In the growth phase the number of cases was growing at an exponential pace and then in containment the number cases per day started to decrease and for that you need substantial reduction in the infection rate. According to the classical SIR epidemiological model the increase in the number of new infections is roughly the product of the infection rate and the number of infected and susceptible ($\alpha \cdot I \cdot S$). If the number of newly infected is not increasing exponentially then this indicates that infection rate has dropped by a magnitude. In other words, in the latter half of the model the epidemic is drastically contained and new generations are not generally being formed over this 28 day period and we are measuring molecular remnants of older generations after the containment. Therefore, our model parameters are comparable to the epidemiological data.

Also, "viral reproductions" looks like it refers to within host replication, so please reword

and clarify the whole discussion lines 166-172.

Please also adjust the claims in the Discussion according to the comments above.

We agree that the viral reproductions could be misinterpreted as within host replication, we replaced the term "viral reproduction" with viral generations.

Minor comments:

line 10: "that" were targeted by the healthcare system

We added "that" to the sentence.

REVIEWERS' COMMENTS

Reviewer #2 (Remarks to the Author):

With the text now submitted, the authors have presented a substantial improvement of their previous manuscript.

By taking into account numerous comments of the reviewers, the presentation of the collected data, their analysis and the authors' interpretations are scientifically clearer and well understandable for the readers.

I have no other significant comments or suggestions for improvement and recommend publication of the manuscript in Nature Communications.

Reviewer #3 (Remarks to the Author):

I appreciate the efforts of the authors to clarify their statements and provide a more nuanced view of their findings.

I should also say that recent evidence (Hay et al 2020, <https://www.medrxiv.org/content/10.1101/2020.10.08.20204222v2>) supports the interpretation of the authors - i.e. that their findings would be naturally explained by changes in epidemiological dynamics.

The paper is well worth publishing, even in the current form.

The only minor comment I have is that the authors could clarify to the reader that most of the effects are likely not due to changes e.g. to the intrinsic generation time that changes across epidemic phases, but to the "observed" one, since the underlying statistical reason is a combination of epidemic dynamics and right censoring. Or, put more clearly, individuals that are more recently infected or have transmitted the virus in a short time are more frequently sampled at any point in time in the fast-growing phase of an epidemic.

We addressed the last comment of the reviewer by adding this sentence to the discussion “In this interpretation we assume that the time within an individual from initial infection to being infectious stays the same throughout the epidemic.